# Long Term Dietary Supplementation with Omega-3 Fatty Acids in Charolais Beef Cattle Reared in Italian Intensive Systems: Nutritional Profile and Fatty Acids Composition of *Longissimus lumborum* Muscle

**DOI:** 10.3390/ani12091123

**Published:** 2022-04-27

**Authors:** Carlo Corino, Francesco Vizzarri, Sabrina Ratti, Mirco Pellizzer, Raffaella Rossi

**Affiliations:** 1Department of Veterinary Medicine and Animal Science, Università degli Studi di Milano, Via Dell’Università 6, 26900 Lodi, Italy; carlo.corino@unimi.it (C.C.); sabrinaratti@hotmail.com (S.R.); 2Department of Agricultural and Environmental Science, University of Bari Aldo Moro, Via G. Amendola 165/A, 70126 Bari, Italy; francesco.vizzarri@uniba.it; 3Veterinary Private Practitioners, 35100 Padova, Italy; pellizzermirco@libero.it

**Keywords:** Charolais beef cattle, cholesterol, fatty acid, lipid supplement, meat quality

## Abstract

**Simple Summary:**

The nutritional quality of meat is a key factor affecting consumer preferences and health. In particular, intramuscular fat content and fatty acid composition, as well as protein, trace minerals and vitamins have a valuable impact on the nutritional quality of meat. The relationship between diet and the occurrence of several diseases in humans has drawn attention to improving the production of meat and meat products, with a high content of omega-3 fatty acids. In fact, over the last decades, a decline in the intake of omega-3 fatty acids has been observed in Western diets, with a growing imbalance between omega-6 and omega-3 fatty acids. Omega-3 enrichment of food is a useful approach to counteract deficiencies of omega-3 fatty acids in human diets. Dietary supplementation of omega-3 fatty acids in ruminants has been studied to positively influence the nutritional quality of the meat, improving consumers’ health. The present study highlights that dietary supplementation of an omega-3 lipid supplement in Charolais beef cattle reared in intensive conditions enhances the meat’s nutritional quality parameters.

**Abstract:**

Recently, the quality of beef has received great attention, and health concerns have been focused on fatty acid composition in relation to dietary requirements. The present work aims to evaluate the effect of omega-3 fatty acids (FA) lipid supplement in beef diet on the nutritional characteristics of *Longissimus lumborum* (LL) muscle. One hundred and eighty Charolais beef were divided in two groups: the control group (CON) received a basal diet and the second one (TR) an isoenergetic diet containing the omega-3 supplement. Dietary treatment did not affect (*p* > 0.05) growth performances, carcass characteristics and LL colour indices. Cholesterol content resulted lower (*p* < 0.001) in LL muscle from TR group than CON. The omega-3 FA and conjugated linoleic acid content were higher (*p* < 0.001) in LL muscle from TR than CON. As expected, LL muscle from TR group showed an increased value of malondialdehyde than CON during refrigerated storage, anyway, remaining within the threshold value of 1 mg/kg meat. In conclusion, the lipid supplement, rich in omega-3 FA improves the fatty acid profile and decreases cholesterol content of LL muscle. This feeding practice is suggested to enhance the nutritional value of meat from beef reared in intensive condition, improving the consumer’s health.

## 1. Introduction

In Italy, 1.056 million beef cattle, representing 47.2% of animals reared for meat production, are imported. In particular, 79.7% of these animals are introduced from France and the main breeds are Charolaise and Limousine. Moreover, in Italy the third most consumed meat is beef with 16.8 kg per capita/year (ISMEA, www.ismeamercati.it (accessed on 24 March 2022).

Beef is an excellent source of proteins with high biological value, vitamins (vitamin B12, B6, riboflavin, thiamine and pantothenic acid), and microelements (zinc, iron, selenium, magnesium, potassium and copper) [1,2]. In addition, beef is also a source of several bioactive compounds such as conjugated linoleic acid (CLA), taurine, creatine, and carnitine [3]. 

The nutritional characteristics of meat are key factors for a proper food choice and a healthier diet [4]. Recently, beef quality has received great attention and the health concerns have been focused on fat content and fatty acid composition in relation to several dietary strategies [5,6]. This aspect has induced changes in consumer demand and has directed producers to modify some meat characteristics through accurate animal nutrition and breeding [6,7,8]. 

The associations between dietary fat and the incidence of several diseases including cardiovascular diseases are clearly recognized [9]. In fact, an inadequate intake of omega-3 polyunsaturated fatty acids (PUFA) in relation to omega-6 PUFA is observed [10]. Diets rich in omega-6 PUFA are linked with inflammation, blood vessel constriction, and platelet aggregation. In contrast, omega-3 PUFA play an important role in human health and are involved in brain and retinal tissue development and in the prevention of inflammatory diseases, cancer, cardiovascular and other chronic diseases [11]. Moreover, conjugated linoleic acid (CLA), naturally produced mainly in ruminant species, has the potential to reduce the risk of cancer, cardiovascular diseases, diabetes, and obesity [12]. 

The need for diets containing high levels of omega-3 PUFA has focused the scientific research on meat as a natural supplier of these healthy substances, enhancing the nutritional quality of meat [13]. Considering the relationship between the fatty acids ingested by animals and those that bypass the rumen intact and will be deposited in the muscle, great attention has been focused on dietary sources able to modify beef fatty acid composition. In fact, the key factor is to identify the correct dietary ingredient or supplement able to increase the deposition of PUFA and omega-3 fatty acids in beef meat. In fact, a previous study reported that the inclusion of linseed increases omega-3 FA in LL of beef [5]. It is also reported that the inclusion of linseed oil or extruded or crushed linseed in beef rations improves muscle fatty acid composition and decreases SFA content and omega-6/omega-3 ratio [14]. Other sources of dietary fatty acids in beef rations, in addition to forage and grass, are plant oils, oilseeds, fish oil and marine algae [14].

Intensive beef cattle farms in Italy are fattening units characterized by the finishing phase of young beef, in particular Charolais, mainly imported from France. Usually, fattening beef are fed corn-based concentrates and corn grain silage with the addition of fodder, typically hay or straw, which are provided as total mixed rations (TMR) [15]. It is reported that beef fed on TMR, composed of corn silage and concentrate, have a higher saturated fatty acid (SFA) and omega-6 PUFA, and lower omega-3 PUFA and CLA, compared with grass- or pasture-fed beef [16].

Therefore, in this breeding condition, nutritional strategies to enhance beef nutritional quality are needed. The present study was performed to assess the effect of dietary lipid supplement, rich in omega-3 PUFA, in Charolais beef cattle on the nutritional characteristics, fatty acid composition, and cholesterol content of the *Longissimus lumborum* (LL) muscle.

## 2. Materials and Methods

### 2.1. Animals and Diets

The animals used in this experiment were reared following the European Union guidelines (2010/63/EU) and approved by the Italian Ministry of Health (D. Lgs. n. 26/2014).

One hundred and eighty Charolais beef cattle (12 ± 1 months of age and 355 ± 35 kg of live weight) imported from France, were bred on a commercial fattening farm located in northern Italy. The animals were grouped by body weight, and randomly assigned to the two experimental groups. Beef cattle were housed in 20 boxes with straw litter (4.6 m^2^ per animal) and were dived into two experimental groups (10 boxes/treatment and 9 beef/box). The control group (CON) received a basal diet containing 80 g/animal/day of calcium soap and the second group (TR) an isoenergetic and isoproteic diet supplemented with 140 g per animal/day of a lipid supplement containing extruded linseed, protected linseed oil and 100 mg/kg of vitamin E (Zoofarma, Villafranca di Verona, Italy). The fatty acid composition of the lipid supplement was total SFA 12.2%, total MUFA 20.1%, and total PUFA 67.9%, (51.0% of C18:3n-3, and 16.7% of C18:2n-6). The total mixed rations were formulated to meet or exceed nutrient requirements for beef according to National Research Institute of Animal Production [17]. The diet ingredients and nutritional composition are reported in Table 1.

Beef finishing systems are generally divided into three phases, with Phase 1 lasting 4 weeks, Phase 2 lasting 12 weeks, and Phase 3 lasting 18 weeks. Animals were fed a TMR composed of corn silage and concentrate and were allowed free access to water and trace-mineralized salt blocks during the experiment. The trial lasted 240 days, until slaughter.

The animals were weighed at arrival and at the end of the experimental trial, and the average daily gain (ADG) and feed conversion ratio (FCR) were calculated.

### 2.2. Carcass Traits

At the end of the trial, the animals were slaughtered, following standard handling procedures at the abattoir (Pantano Spa, Arre, Italy) at about 19 months of age. Live weight at slaughter and hot carcass weight were recorded and dressing percentage (hot carcass weight/live weight at slaughter) was calculated. Carcasses were stored at 2 °C for 24 h, then the left LL muscle was randomly selected from 20 beef per treatment (2 beef/box) and excised from each carcass at the seventh and eighth lumbar vertebra. The ice-cooled samples were transported to lab of the Department of Veterinary Medicine and Animal Science for the meat quality determination.

### 2.3. Physical Parameters

Analytical determinations were carried out immediately after the arrival at the laboratory. The colour indexes were measured on the samples at days 0, 4, 7 and 10 of storage at 4 °C, using a CR-300 ChromaMeter (Minolta Camera Co., Osaka, Japan). The instrument was calibrated on the CIE LAB colour space system employing a white calibration plate (Calibration Plate CR-A43, Minolta Cameras). The colorimeter had an 8 mm measuring area and was illuminated with a pulsed Xenon arc lamp (illuminated C) at a 0° viewing angle. Reflectance measurements were obtained at a viewing angle of 0° and the spectral component was included. Each data is the mean of six replications at the chop surface.

### 2.4. Chemical Parameters

Determinations of moisture (method 985.41), ash (method 920.153), fat (method 180 960.39) and crude protein (method 928.08) content were performed in duplicate on the samples of LL according to the methods of the Association of Analytical Chemists [18].

### 2.5. Cholesterol Content

The cholesterol content was determined using the method of Maraschiello et al. [19] and measured by HPLC. For the analyses a Kontron HPLC (model 535, Kontron Instruments, Milan, Italy) equipped with a C18 reverse-phase column (250 mm × 4.6 mm × 5 µm; Phenomenex, Torrance, CA, USA) was employed. The mobile phase used was acetonitrile: 2-propanol (55:45 *vol*/*vol*) at a flow rate of 1.2 mL/min. The detection wavelength was 210 nm and the retention time was 13.89 min. The quantitative determination of the cholesterol content was based on an external standard method, using a pure cholesterol reference standard (Sigma, St. Louis, MO, USA).

### 2.6. Fatty Acid Composition

The lipids from intramuscular fat were extracted by a modified method of Folch et al. [20]. Samples (15.00 ± 0.01 g) were homogenized for 2 min with 90 mL of chloroform–methanol solution (2:1 *v*/*v*). Then, 30 mL of chloroform and 30 mL of deionized water were incorporated, and the mix was homogenized. A 0.58% aqueous NaCl solution was added to the homogenate, causing the chloroform layer (containing lipid) to separate from the methanol–water phase. The lipid extract was moved to a 250 mL flask and the solvent evaporated under a stream of nitrogen. The lipid content was determined gravimetrically. Fatty acids were transesterified by sodium methoxide in methanol [21]. Fatty acids were quantified using a gas chromatograph Trace GC-2000 series equipped with a flame ionization detector (ThermoQuest, Mississauga, ON, Canada; CE, Instruments, Milan, Italy) with a fused silica capillary column (SP-2560, 100 m × 0.25 mm; Supelco Inc., Bellefonte, PA, USA). Helium was the carrier gas with the injector temperature set at 250 °C and the detector temperature at 270 °C. The temperature of the oven was programmed as follows: 70 °C for 2 min increased by 3 °C/min to 100 °C, held for 5 min, increased at 3 °C/min to 165 °C, held for 10 min, increased at 3 °C/min to 220 °C and held for 35 min. The total run time was 102 min. Individual fatty acid methyl esters were identified using a standard (Supelco TM 28 FAME Mix components, Bellefonte, PA, USA) and quantified using the internal standard methyl-nonadecanoic acid ester (Sigma Chemical Co., St. Louis, MO, USA). Moreover, individual CLA cis-9, trans-11 (Sigma Chemical Co., St. Louis, MO, USA) was used to identify the CLA isomer.

The health lipid indices were estimated used the Ulbricht and Southgate method [22]. Indices of atherogenicity (IA) and thrombogenicity (IT) were calculated by the following formulas:IA = [(4 × 14:0) + (16:0)] × [PUFA (n6 and n3) + MUFA)] ^−1^
(1)
IT = [(14:0) + (16:0) + (18:0)] × [(0.5 × MUFA) + (0.5 × n6) + (3 × n3) + (n3 × n6 − 1)] ^−1^. (2)

### 2.7. Oxidative Stability

Lipid oxidation in relation to storage time (0, 4, 7 and 10 days) at 4 °C was determined by the thiobarbituric acid reactive substances (TBARS) method of Realin et al. [23]. All the analyses were performed in duplicate. The absorbance at 532 nm was measured with Varian Cary 100 UV-VIS spectrophotometer (Varian, Belrose, Australia). The TBARS value, expressed as the mg malonaldehyde/kg meat, was obtained using a conversion factor based on a standard curve using 1,1,3,3-tetraethoxypropane (TEP; Sigma-Aldrich, Milan, Italy).

### 2.8. Statistical Analyses

Statistical analyses of the data were performed using SPSS (SPSS/24 PC Statistics 26.0 IBM, New York, NY, USA). The data on growth performances and meat quality parameters were analysed taking into consideration diet as the main effect. Means were compared using Student’s t test. The data on colour parameters and oxidative stability were analysed by repeated measure ANOVA to assess the main effect of treatment and time and their interaction. Box was considered as the experimental unit for growth performances. Beef was considered as the experimental unit for meat quality parameters. Data are presented as means ± SEM, and a value of *p* < 0.05 was used to indicate statistical significance.

## 3. Results

### 3.1. Growth Performance and Carcass Characteristic

No effect of dietary treatment (*p* > 0.05) was observed on growth performances and carcass characteristics. The average final weight of beef was 717 ± 37 kg, and the ADG results were 1.47 ± 0.4 kg/d in the control group and 1.58 ± 0.3 kg/d in the group fed the lipid supplement. Also, the feed conversion ratio was unaffected (*p* > 0.05) by dietary treatment (6.34 ± 0.50 kg/kg in CON and 6.82 ± 0.54 kg/kg in TR group).

The average carcass weight was 451.5 ± 28.6 kg and was not affected (*p* > 0.05) by dietary treatment. The dressing percentage was 62.8 ± 3.0 in the control group and 62.3 ± 2.5 in the TR group.

### 3.2. Colour Indexes

The changes in LL muscle colour indices, lightness (L*), redness (a*) and yellowness (b*), in relation to dietary treatment and storage time at 4 °C are shown in Figure 1A–C respectively.

The lightness values were significantly affected by storage time (*p* < 0.001) but not by dietary treatment (*p* > 0.05). No interaction between treatment and time was observed (*p* > 0.05). The redness values were significantly affected by storage time (*p* < 0.001). No treatment effect or interaction between time and treatment was observed (*p* > 0.05). The yellowness values were significantly affected by storage time (*p* < 0.001). No treatment effect or interaction between time and treatment was observed (*p* > 0.05).

### 3.3. Chemical Parameters

The chemical composition of the LL muscle is reported in Table 2. The chemical composition of the LL muscle did not differ (*p* > 0.05) for dry matter, protein, and fat content. The LL samples from CON group had a higher ash content (*p* < 0.05) compared with controls.

The LL muscle from beef fed the lipid supplement showed a lower cholesterol value compared to controls (54.62 mg/100g CTR vs. 39.45 mg/100g TR; *p* < 0.001) as reported in Figure 2. In fact, the cholesterol content of the LL muscle of the TR group is 38% lower than that of the controls. Our data showed that dietary supplementation with omega-3 fatty acid, in an Italian intensive beef system, can decrease the LL cholesterol content.

### 3.4. Fatty Acid Composition

Fatty acid composition of LL muscle in relation to dietary treatment are reported in Table 3. The LL muscle of the animals from the TR group contained a higher content of MUFA and PUFA (*p* < 0.001) than the controls. A higher content of *trans* vaccenic acid (C18:1 *trans*-11) and CLA (C18:2 *cis*-9, *trans*-11) was observed in LL muscle from TR group than from controls.

The PUFA to SFA ratio resulted higher (*p* < 0.001) in the TR group than the control group. Therefore, a lower atherogenic and thrombogenic index was observed in the TR group than in the control group.

As reported in Figure 3 a higher content of omega-3 (+249.2% *p* < 0.001) and omega-6 (+102.1%; *p* < 0.001) FA was observed in the LL muscle of cattle fed a fat supplement diet. The omega-6 to omega-3 ratio was lower (−47.2%; *p* < 0.001) in the TR group than in the control group.

### 3.5. Oxidative Stability

The oxidative stability of the LL muscle in relation to dietary treatments and storage time is reported in Figure 4. As expected, the TBARS values were significantly affected (*p* < 0.05) by dietary treatment, with a lower value in the CON group than in the TR group. The TBARS content in the LL muscle of the CON group increased from 0.13 mg/kg (initial value) to 0.55 mg/kg at 10 d of storage. In the samples from the TR group the TBARS content increased from 0.24 mg/ kg (initial value) to 0.74 mg/kg at 10 d of storage. The time of storage (from 0 d to 10 d) determined a significant increase (*p* < 0.001) in the malondialdehyde (MDA) content in all meat samples. No significant interaction between storage time and dietary treatment was observed (*p* > 0.05).

## 4. Discussion

In the present study, no effect of dietary treatment on growth performances and carcass characteristics was observed, in agreement with previous studies on beef fed extruded linseed [24,25,26]. Also, Baltrukonienė et al. [27] reported that dietary integration with rapeseed and linseed cake, containing omega-3 fatty acids, did not affect beef slaughter weight, carcass weight or dressing percentage. In contrast, a reduction in dressing percentage was observed in beef fed 5% whole linseeds [28]. The lack of effect of lipid supplement containing omega-3 on beef performance may be due to the to the comparable dry matter intake of isoenergetic diets.

Beef LL colour parameters during the storage period indicated that colour was influenced by the storage time, but not by dietary treatments. Our data are in agreement with Mach et al. [29] and Conte et al. [30] who reported that linseed dietary supplementation in beef did not affect the meat colour indices.

Previous studies reported that storage time significantly affects colour parameters in beef. In fact, the lightness (L*) values decreased (*p* < 0.05) in both experimental groups from seven days of storage. A decrease in redness values (a*) in LL beef muscle was observed [31] and the trends reported are comparable with our data. This phenomena in red meat is usually related with oxidation of the iron atom within the heme group in red oxymyoglobin to brownish metamyoglobin, and it is also associated with lipid oxidation [32].

Results of the current study support that dietary supplementation, with fat supplement containing omega-3 fatty acids, had no detrimental effects on meat colour during refrigerated storage. This is positive considering that colour perception plays a central role in the consumer’s purchase decision and the length of aging for these types of meat [33,34].

Dietary treatment did not affect LL muscle dry matter, protein or fat content. These data agree with Bartoň et al. [35] in Charolais heifer fed a diet containing extruded linseed. Further, Mapiye et al. [5] reported no difference in muscle chemical composition of beef fed with flaxseed.

The cholesterol content was lower in LL muscle from the TR group, with a positive effect on meat nutritional quality. Our data are in agreement with Bartoň et al. [35] who reported the same cholesterol content in the LL muscle of Charolais beef. In disagreement with our data, an increase in cholesterol content was observed in LL from Charolaise × Podolica young bulls fed with extruded linseed [25]. Moreover, feeding extruded linseed in Maremmana young bulls did not affected the cholesterol meat content [30].

The recommended maximum daily intake (RMDI) of cholesterol is set at 300 mg/day [36]. Therefore, the consumption of an average portion of 200 g of LL muscle from CON or TR beef represents a cholesterol intake of 109.24 and 78.9 mg respectively, corresponding to 36.4% and 26.3% of RMDI.

The present data on LL muscle FA composition highlights the beneficial effects of dietary integration with omega-3 fat supplementation in beef. In developed countries, consumers are encouraged to reduce the dietary intake of SFA, increasing PUFA intake [37]. The PUFA/SFA and omega-6/omega-3 ratios are important for defining the nutritional value of meat. The nutritional recommendations suggest that the ratio of PUFA:SFA should be above 0.4 and the ratio of omega-6/omega-3 PUFA should be less than 4 [38].

The PUFA/SFA ratio for beef is typically low at around 0.1 [39]. In the present study, it was better in the TR group (0.16) than CON (0.06) (*p* < 0.001). As reported, highly saturated fat is considered detrimental for human health [40]. Similar values, ranging from 0.07 to 0.18, were observed in the literature [41,42].

The AI and TI indices characterize the atherogenic and thrombogenic potential of the diet [22]. It is assumed that AI below 1 is beneficial for human health [42]. In the present study the AI of the TR group are lower than this threshold value (Table 3). In addition, TI was lower in LL muscle from TR groups than from control groups due to a low SFA content, especially palmitic (C16:0) and stearic fatty acid (C18:0), and a high content of omega-3 PUFA.

The omega-6/omega-3 ratio in the human diet is assumed to be more relevant than the PUFA/SFA ratio [43]. Its balance plays an important role in the prevention of several chronic and autoimmune diseases. In our study, the omega-3 content in LL muscle was 41.17 mg/100 g in CON group and 83.23 mg/100 g in TR group. The omega-6 content in LL muscle was 19.34 mg/100 g in the CON group and 67.65 mg/100 g in the TR group. Therefore, the omega-6/omega-3 ratio was lower in LL muscle from the TR group (1.25) than from that obtained from the controls (2.36). As reported in the literature, the meat content of omega-6 PUFA and omega-3 PUFA in intensive production conditions range from 1.5–8%, and 0.30–2.8%, respectively [44,45]. In fact, beef fed corn silages have a high omega-6/omega-3 ratio due to the high linoleic acid (C18:2n-6) content in corn [46]. Dietary integration with lipid supplement, containing omega-3 PUFA, can reduce this ratio, increasing the healthiness of the LL muscle fatty acid profile.

Moreover, human intake of long chain omega-3 fatty acids such as eicosapentaenoic acid (EPA, 20:5n-3) and docosahexaenoic acid (DHA, 22:6n-3), have been linked with a diminished risk of cardiovascular disease and cognitive decline [47]. It is estimated that less than 20% of the population consumes >250 mg/day of seafood omega-3 PUFA [48]. In the present study an increase of EPA, but not of DHA content, was observed in LL muscle from beef fed the lipid supplement, in agreement with literature. In fact, several studies report that diets rich in linolenic acid (C18:3n-3; ALA) increased levels of EPA with no effect on DHA level [5,49].

Beef products are also a dietary source of CLA [50]. The predominant CLA in beef is the *cis*-9, *trans*-11 isomer, which has several health promoting properties including antitumoral and anticarcinogenic activities [51]. The *cis*-9, *trans*-11 CLA is produced from two dietary sources, one originates from ruminal biohydrogenation of linoleic acid to stearic acid in the rumen by *Butyrinvibrio fibrisolvens* and other bacteria and the second one is the endogenous conversion of *trans* vaccenic acid (C18:1 *trans* 11) by Δ9-desaturase in ruminant adipose tissue [52]. The *trans* vaccenic acid is an intermediate produced during the ruminal biohydrogenation of ALA and linoleic acid (C18:2 n6). Fresh grass, grass silage, and pasture feeding during the finishing period, resulted in increased deposition of *cis*-9, *trans*-11 CLA in muscle [16,53]. In intensive conditions, the dietary content of ALA is very low because the beef ration contained corn silage and concentrate [39]. In the present study, dietary integration with omega-3 supplementation resulted in a key factor to increase *trans* vaccenic acid (22.66 mg/100 g meat CTR vs. 47.44 mg/100 g meat TR) and CLA (5.76 mg/100 g meat CTR vs. 24.97 mg/100 g meat TR) content in beef LL muscle.

Moreover, it is also reported that linseed supplementation to finishing beef in intensive conditions enhances the juiciness and tenderness of the meat compared to meat from beef fed a corn-based diet [54]. Other studies reported a lower meat sensory attribute due to beef dietary supplementation with omega-3 in beef [55] and pork [56], but this is generally related to the low antioxidant content.

Previous studies highlight that dietary strategies, to improve omega-3 fatty acids content in meat, also required an increased level of antioxidants to protect the meat from oxidative processes and off-flavour development [26,57].

An increased consumer preference for omega-3 enriched beef is also observed [58]. Consequently, improving the omega-3 fatty acids content in beef muscle offers the consumers a meat that is closer to the nutritional recommendations for a healthy diet (omega-3 long chain PUFA adequate intake comprised from 0.25 to 1 g/d), positively affecting purchasing choices [36]. Thus, the consumption of an average portion of 200 g of LL muscle from CON or TR beef provides 12.7 and 56.6 mg omega-3 long chain PUFA respectively, corresponding to 5.1% and 22.6% of RMDI.

As expected, dietary supplementation with omega-3 lipid supplement decreased oxidative stability in the LL muscle, due the higher presence of unsaturated fatty acids [59]. The oxidation processes of lipid affect the meat quality of beef during processing and storage. Generally, oxidized meat presents a low nutritional and sensory quality and there is also the possible development of toxic compounds that negatively affect consumer health [60]. In the present study the TBARS values in both the experimental groups remained lower than the threshold value at about 1 mg/kg meat [61].

## 5. Conclusions

In conclusion, the present data show that long term dietary supplementation with omega-3 fatty acids in Charolais beef, reared in Italian intensive conditions, is a valid strategy to improve the healthy value of beef meat. Dietary supplementation in *in field* conditions did not affect growth performances and carcass characteristics. The improvement in the fatty acid profile, and the lower cholesterol content together with the MDA value lower than the threshold, makes this meat healthy for consumers. Further studies are needed to evaluate the effects of the lipid supplement on the technological and sensory parameters of meat. This feeding practice is suggested to enhance the nutritional and health value of meat from beef reared in intensive condition.

## Figures and Tables

**Figure 1 animals-12-01123-f001:**
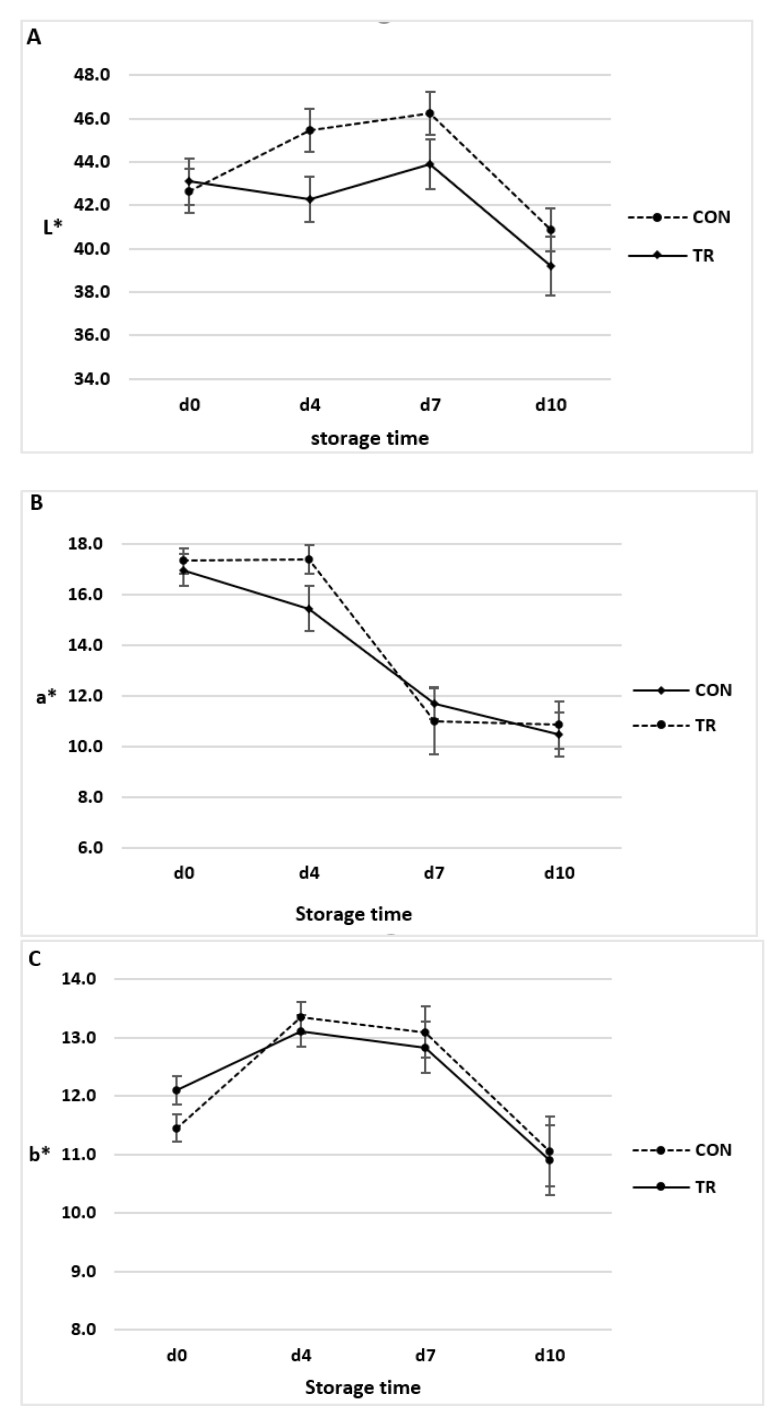
Colour indices of *Longissimus lumborum* of Charolais beef cattle fed control diet (CON) and diet supplemented with omega-3 lipid supplement (TR) in relation to storage time. *n* = 10; data are reported as mean ± SEM. (**A**) Lightness (L*) values: effects of treatment, *p* = 0.140; time, *p* < 0.001; time xtreatment, *p* = 0.109; (**B**) Redness (a*) values: effects of treatment, *p* = 0.648; time, *p* < 0.001; time x treatment, *p* = 0.570; (**C**)Yellowness (b*) values: effects of treatment, *p* = 0.994; time, *p* < 0.001; time x treatment, *p* = 0.734.

**Figure 2 animals-12-01123-f002:**
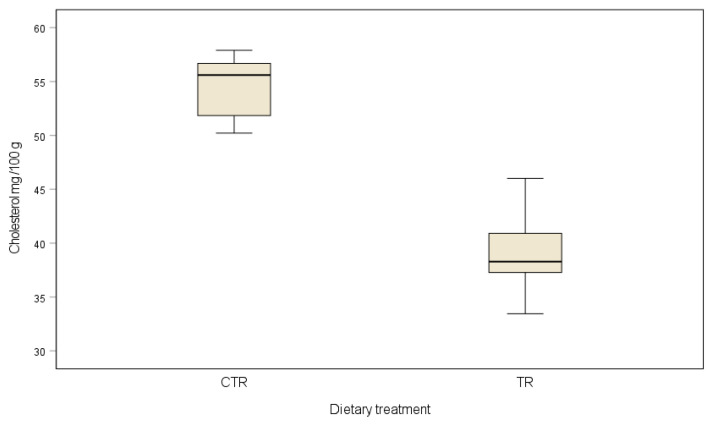
Spider plot of *Longissimus lumborum* muscle cholesterol content of Charolais beef cattle fed control diet (CON) and diet supplemented with omega-3 lipid supplement (TR).

**Figure 3 animals-12-01123-f003:**
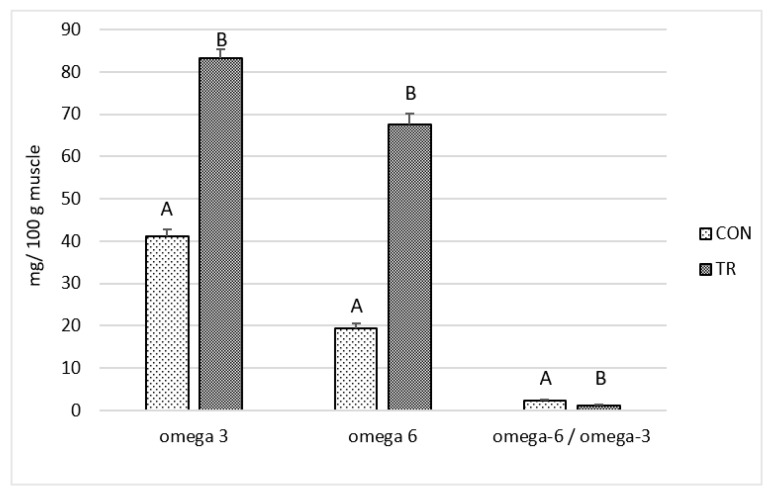
Omega-6 and omega-3 fatty acids (mg/100 g muscle) in *Longissimus lumborum* muscle of Charolais beef cattle fed a control diet (CON) and a diet supplemented with omega-3 lipid supplement (TR). *n* = 10; data are reported as mean ± SEM. ^A,B^ for *p* < 0.001.

**Figure 4 animals-12-01123-f004:**
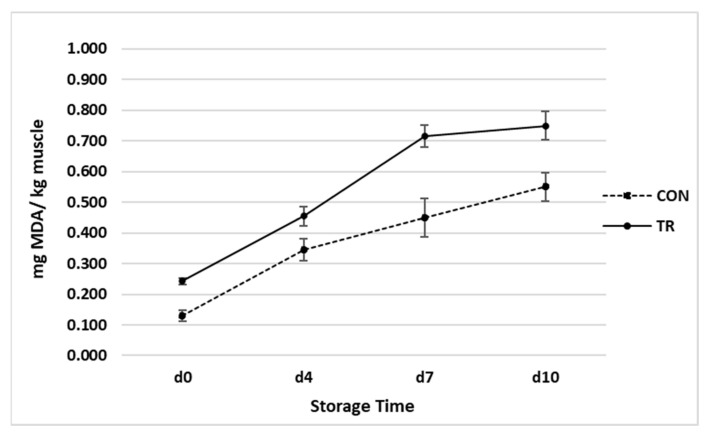
Oxidative stability of *Longissimus lumborum* of Charolais beef cattle fed a control diet (CON) and a diet supplemented with omega-3 lipid supplement (TR) in relation to storage time. *n* = 10; data are reported as mean ± SEM. Effects of treatment, *p* = 0.002; time, *p* < 0.001; time ∗ treatment, *p* = 0.363.

**Table 1 animals-12-01123-t001:** Ingredients and nutritional composition of control diet (CON) and diet supplemented with omega-3 lipid supplement (TR) during different fattening phases.

Total Mixed Ration	Phase 1		Phase 2		Phase 3	
	CON	TR	CON	TR	CON	TR
Ingredient. % D.M.						
Corn silage	19.15	19.15	23.72	23.72	19.73	19.73
High moisture corn	10.11	10.11	15.96	15.96	21.27	21.27
Beet pulp	22.91	22.91	9.84	9.84	10.64	10.64
Wheat straw	18.33	18.33	11.80	11.80	9.82	9.82
Distillers	9.40	9.40	6.78	6.78	8.09	8.09
Soybean meal	6.11	5.64	11.37	10.93	8.73	8.36
Corn meal	5.05	5.05	12.24	12.24	14.82	14.82
Wheat middlings	5.17	5.17	4.81	4.81	4.00	4.00
Premix ^1^	2.82	2.82	2.62	2.62	2.18	2.18
Calcium soap	0.94	-	0.87	-	0.73	-
Lipid supplement	-	1.41	-	1.31	-	1.09
Nutrient levels % DM						
Energy, UFC/kg	0.89	0.89	0.96	0.96	0.98	0.98
CP	13.84	13.98	14.3	14.2	13.6	13.7
NDF	41.9	42.8	33.6	34.4	31.6	32.3
Starch	20.1	20.2	29.4	30.9	32.9	33.4
EE	2.99	3.07	3.4	3.4	3.5	3.4
Ca	0.96	0.95	0.80	0.80	0.65	0.67
P	0.30	0.28	0.26	0.26	0.25	0.25

^1^ Provided per kg: CaCO_3_ 150 g, NaHCO_3_ 28 g, NaCl 10 g, MgO 15 g, Vitamin A 2,000,000 IU; Vitamin D3 17,500 IU; Vitamin E 300 mg; Vitamin B1 65 mg; Vitamin B2 25 mg; Vitamin B6 10 mg; choline 4000 mg; CoCO_3_ 67 mg; CuSO_4_ 315 mg; Mn_2_O_3_ 6450 mg; Na_2_SeO_3_ 33 mg; ZnSO₄ 16,500 mg; FeCO_3_ 8300 mg; KI 21 mg, urea 70,000.

**Table 2 animals-12-01123-t002:** Chemical characteristics of *Longissumus dorsi* muscle of Charolais beef cattle fed control diet (CON) and diet supplemented with omega-3 lipid supplement (TR).

Item ^1^	CON	TR	*p*-Value
Moisture %	73.78 ± 0.19	73.48 ± 0.26	0.940
Crude protein, % ^2^	22.78 ± 0.26	23.17 ± 0.19	0.186
Crude fat, % ^2^	1.83 ± 0.12	2.17 ± 0.19	0.254
Ash, % ^2^	1.17 ± 0.03 ^a^	1.09 ± 0.01 ^b^	0.031

^1^ Data are reported as mean values ± SEM, *n* = 10; ^2^ Data expressed as percentage of wet weight; ^a,b^ on the same row differed for *p* < 0.05.

**Table 3 animals-12-01123-t003:** Fatty acid profile (mg/100 g muscle) of *Longissimus lumborum* muscle of Charolais beef cattle fed a control diet (CON) and a diet supplemented with omega-3 lipid supplement (TR).

Fatty Acid Composition	CON	TR	*p*-Value
C14:0	74.02 ± 0.97	75.74 ± 2.53	0.160
C16:0	633.84 ± 5.42	663.07 ± 8.68	<0.05
C16:1	7.20 ± 0.41	14.18 ± 0.60	<0.001
C17:0	20.04 ± 0.96	18.38 ± 1.27	0.103
C18:0	323.54 ± 3.00	318.07 ± 6.12	0.142
C18:1 *trans* 11	22.66 ± 1.16	47.44 ± 0.83	<0.001
C18:1 *cis* 9	596.57 ± 4.73	760.12 ± 6.95	<0.001
C18:2 n6	34.36 ± 1.16	59.76 ± 2.21	<0.001
C18:2 *cis* 9, *trans* 11	5.76 ± 0.76	24.97 ± 0.25	<0.001
C18:3 n3	14.02 ± 1.16	54.93 ± 1.49	<0.001
C20:0	69.39 ± 2.24	54.94 ± 1.49	<0.05
C20:4 n6	6.81 ± 0.74	23.48 ± 0.13	<0.001
C20:5 n3	5.12 ± 0.38	12.40 ± 1.30	<0.001
C22:0	0.57 ± 0.08	0.76 ± 0.12	0.125
C22:5 n3	3.59 ± 0.21	6.34 ± 0.56	<0.001
C22:6 n3	0.35 ± 0.04	0.56 ± 0.09	0.476
SFA ^1^	1120.87 ± 6.20	1128.93 ± 7.17	0.201
MUFA ^2^	626.43 ± 4.85	821.74 ± 6.37	<0.001
PUFA ^3^	70.01 ± 1.88	182.43 ± 3.69	<0.001
PUFA/SFA	0.06 ± 0.002	0.16 ± 0.004	<0.001
AI ^4^	1.36 ± 0.021	0.96 ± 0.010	<0.001
TI ^5^	2.54 ± 0.046	1.53 ± 0.011	<0.001

*n* = 10; data are reported as mean ± SEM. ^1^ SFA, saturated fatty acids. ^2^ MUFA, monounsaturated fatty acids. ^3^ PUFA, polyunsaturated fatty acids. ^4^ AI, atherogenicity index. ^5^ TI, thrombogenicity index.

## Data Availability

The data presented in this study are available on request from the corresponding author.

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
