# Peer review of "Long Term Dietary Supplementation with Omega-3 Fatty Acids in Charolais Beef Cattle Reared in Italian Intensive Systems: Nutritional Profile and Fatty Acids Composition of Longissimus lumborum Muscle"

_animals, 2022, doi:10.3390/ani12091123_

Round 1

Reviewer 1 Report

Minor revision of manuscript is needed. My comments are shown in the attached file.

Reviewer 2 Report

Revision: 11.04.2022

Long term dietary supplementation with Omega-3 fatty acids  in Charolais beef cattle reared in Italian Intensive System:  nutritional profile and fatty acids composition of Longissimus Lumborum muscle

Topic of manuscript is interesting and valuable regarding excessive consumption of saturated fatty acids, especially palmitic ones. Experiment with rearing animals is properly designated. Also the scheme of analyses was properly conducted. However,  I have doubts regarding interpretation of obtained results of fatty acids (FAs) assayed in lipids extracted from meat. Also I found some shortcomings, and doubts which should be revised or/and clarified during the review. My general and specific comments are given below:

General comments:

  1. As can be seen from the description in the manuscript (methodology), Table 3 shows the contents of fatty acids determined in the fat extracted from meat - Longissimus dorsi muscle (expressed in g/100g FAs). Authors also assayed lipid content in this cut of beef carcase.  However there is a lack of the contents of FAs. Many papers are devoted to FA contents of beef meat (mainly regarding feeding), the results being expressed as lipid fraction. Yet, from the consumer’s point of view and for nutritional reasons it is important to present FA contents per 100 g of meat.

Thus I suggest to compute and present the content at least more important FAs and total contents in separate table. You can find such interpretation in rare publications (also with that of rabbit meat).

Such presentation of data improve manuscript and make it more original among other similar publications.

  1. In the whole text should be included information about the content of FAs in fat extracted from LL muscle. You not present contents of FAs in meat. Please revise (also in abstract).

  1. Regarding Introduction part - hypothesis:

 As an supplement in feeding beefs linseed oil (protected with extruded linseed) was applied, why ? Please explain why did you apply this supplement. Please insert some information about previous (published) results experiments with using linseed (if there were any). The hypothesis is not sufficiently substantiated.

Detailed comments:

  1. Please ensure: the trial lasted 240 days, are you sure that the name „long term …..” in title is proper ? (it is just a question). I'm wondering.
  2. Regarding 2.3. in my opinion, proximate composition of meat was analyzed and measurement of colour indices, please separate 2.3 part on two subchapters.
  3. Regarding assay of cholesterol, content please supply more information about quantitative analysis.
  4. Please verify table 2 caption and revise.
  5. I suggest not use term ethyl extract it should by “lipids” or “fat”.
  6. Lines 235-236: the sentence needs revision please correct.
  7. Lines 291-296, please include information about content of cholesterol assayed in the study - 55 mg/mg ?
  8. Regarding Line 316, information about the content of omega 3 and omega 6 expressed on 100g of meat should be supplemented.
  9. Also authors should calculated CLA and vaccenic acid content in mg on 100 g of meat.
  10. There is a lack in conclusion information how 100g (or 200g )of meat would cover requirement for PUFA (for example according to EFSA recommendations), pleasesupply.

Round 2

Reviewer 2 Report

I found that manuscript was sufficiently revised. I also appreciate answers on all comments.